# System Approaches to Water, Sanitation, and Hygiene: A Systematic Literature Review

**DOI:** 10.3390/ijerph17030702

**Published:** 2020-01-21

**Authors:** Nicholas Valcourt, Amy Javernick-Will, Jeffrey Walters, Karl Linden

**Affiliations:** 1Department of Civil, Environmental, and Architectural Engineering, University of Colorado Boulder, Boulder, CO 80309, USA; amy.javernick@colorado.edu (A.J.-W.); Karl.Linden@colorado.edu (K.L.); 2USAID Sustainable WASH Systems Learning Partnership, United State Agency for International Development, Washington, DC 20004, USA; jwalters@georgefox.edu; 3College of Engineering, George Fox University, Newberg, OR 97132, USA

**Keywords:** WASH, systems approaches, systematic literature review, grey literature

## Abstract

Endemic issues of sustainability in the water, sanitation, and hygiene (WASH) sector have led to the rapid expansion of ‘system approaches’ for assessing the multitude of interconnected factors that affect WASH outcomes. However, the sector lacks a systematic analysis and characterization of the knowledge base for systems approaches, in particular how and where they are being implemented and what outcomes have resulted from their application. To address this need, we conducted a wide-ranging systematic literature review of systems approaches for WASH across peer-reviewed, grey, and organizational literature. Our results show a myriad of methods, scopes, and applications within the sector, but an inadequate level of information in the literature to evaluate the utility and efficacy of systems approaches for improving WASH service sustainability. Based on this analysis, we propose four recommendations for improving the evidence base including: diversifying methods that explicitly evaluate interconnections between factors within WASH systems; expanding geopolitical applications; improving reporting on resources required to implement given approaches; and enhancing documentation of effects of systems approaches on WASH services. Overall, these findings provide a robust survey of the existing landscape of systems approaches for WASH and propose a path for future research in this emerging field.

## 1. Introduction

Despite substantial gains in access to water, sanitation, and hygiene (WASH) services over the past thirty years, it is estimated that 2.3 billion people worldwide still lack access to these basic human necessities [1]. For those who have gained access, monitoring trends show that over time, WASH services consistently fail to function as intended [2,3]. This service sustainability issue persists despite a highly uniform and proven set of technologies and approaches that have been developed over the decades within the WASH sector [4]. In Sub-Saharan African countries alone, up to 70% of rural water schemes are estimated to be non-functional or intermittently functional at any given time [5]. Worldwide, low service sustainability contributes to nearly two million preventable deaths and 82 million disability-adjusted life years (DALYs) each year attributed to WASH [6]. These impacts disproportionately affect communities that are rural, poor, and resource-limited [7]. Accordingly, service sustainability is identified as a key challenge to be addressed in order for investments in WASH hardware and software to deliver their intended public health impacts [8,9].

Within the WASH sector there is a growing acknowledgement that existing approaches, which have traditionally focused on the installation and maintenance of hardware (e.g., hand pumps, latrines, hand-washing stations) and community-based management models [10,11,12], will not be sufficient on their own to meet the universal access and service targets of Sustainable Development Goal (SDG) #6: ensuring availability and sustainable management of water and sanitation for all [13,14,15,16,17,18]. Given the interconnected nature of financial, institutional, environmental, technological, and social factors that influence sustained service delivery [19], the sector has started to advocate for a ‘systems’ approach to sustain WASH service delivery [20,21,22]. Central to this approach is the idea that challenges to the sustainability of WASH services are not due to the weakness of any single factor (e.g., cost recovery), but rather the collective effect of a wide range of interacting factors. Thus, improving the sustainability of WASH services requires an enhanced understanding of the combined effect that a multitude of factors exert service delivery outcomes [23,24,25].

This systems perspective of WASH is embraced by many in the sector and has led to a rapid expansion in the number of tools, frameworks, and approaches for understanding the interconnected factors of WASH services [26] including collective action coalitions [27], multi-criteria decision analysis [28], market-based approaches [29], soft systems methodologies [30], composite scoring [31], and system dynamics [32], among others. Indeed a recently published book focused solely on presenting systems approaches to WASH for practitioners [33]. However, to better understand the systems approaches being applied in WASH, there is a need to characterize and synthesize the literature, including the methods, approaches, and applications used. While others have conducted reviews of systems approaches for international development at large [34,35,36], and within the WASH sector in particular [21,37], these studies were not intended to be either comprehensive or systematic in nature, and instead focused on a particular method, approach or application. This lack of understanding of the existing landscape of different methods for systems approaches for WASH may ultimately restrict their use and uptake across the sector, especially for local practitioners who have limited resources to implement complex approaches.

### Study Objectives

To address this need, we seek to characterize the breadth of methods employed for WASH systems approaches, the use of these methods in WASH projects, the impacts these approaches have on service outcomes, as well as identify key knowledge gaps in the existing knowledge base. For the purposes of this study, we define a WASH system as a collection of all of the factors and their interactions which influence WASH service delivery within a given contextual, institutional or geopolitical boundary. We conceptualize factors as any tangible or abstract element, aspect or component thought to directly or indirectly influence the WASH system. Examples include finances, hardware, actors, gender, and socio-economic conditions, among others.

To address these objectives, this study proposes four research questions (RQs): (RQ 1) What methods (i.e., analyses) are being employed for understanding and engaging with WASH systems?; (RQ 2) In what contexts (i.e., geographic and programmatic) are these approaches being applied?; (RQ 3) What evidence exists that these systems approaches improve the sustainability of WASH services?; (RQ 4) What gaps exist in the current knowledge base for systems approaches to WASH?

## 2. Materials and Methods

To address the study objectives, we conducted a systematic literature review based on best practices for research synthesis by Cooper et al. [38] and the Preferred Reporting Items for Systematic Literature Reviews and Meta-Analyses (PRISMA) statement to structure the review [39,40]. The PRISMA checklist used for this review is included in Appendix A. Due to the nature of the study, we did not seek to include an assessment of effects and effect sizes or the quality of evidence.

### 2.1. Literature Identification

We identified a diverse set of 19 databases containing peer-reviewed, grey, and organizational literature on WASH systems approaches. We use the term organizational to refer to literature written and published directly by an organization (e.g., United Nations International Children’s Emergency Fund (UNICEF)); in contrast, we use grey to refer to any literature that is not peer-reviewed and does not represent the work of an individual organization (e.g., books, compendiums, reports). A description of the databases and rationale for inclusion is presented in (Appendix A). A preliminary review of WASH systems literature and past assessments of systems tools within WASH [41,42] revealed that much of the information about systems approaches to WASH existed in grey literature sources and organizations’ repositories, outside of a peer-reviewed journal database. To ensure that applicable literature from grey and organizational sources were included, additional records were obtained from bibliographic hand searches and expert consultations. This included reviewing works cited in applicable literature, conference proceedings, newsletters, and blogs related to systems approaches for WASH.

Based on pilot search results, we noted ambiguity around what constitutes a ‘systems approach’ for WASH; thus, searching for records based on this term alone, or using it as a screening criterion, would not have represented the full breath of literature relevant to our review. To address this, we developed a more inclusive search strategy that contained two primary elements: keywords for WASH and systems. We reviewed indices and key terms from seminal texts on systems-thinking topics to identify keywords. This produced a list of over 400 keywords which were affinity grouped into a representative list of 18 systems keywords (Appendix A) based on comparable terms and frequency of terms. Combining these with select keywords for WASH and select wildcard operators to allow for similar versions of the term, generated the search string formula used in the literature search. Search queries were time-limited to only return documents published after 2000 (the year the Millennium Development Goals (MDGs) were adopted) and before April 2019. Database searches were conducted in August 2018 and repeated in March 2019.

### 2.2. Selection of Articles and Data Extraction

To characterize the breadth of methods and approaches for evaluating WASH systems, a two-step screening process was developed. In the initial screening, papers were vetted for their applicability to WASH and WASH systems. In the secondary screening, studies were reviewed for three attributes—method, scope and application—that were developed to directly address the review’s objectives (RQ1–3, respectively).

#### 2.2.1. Initial Screening

For consistency, the search strategy was implemented across all databases and extracted titles, abstracts, and identifier (Digital Object Identifier (DOI) or other) for each study. All studies were then given a new unique identifier based on the databases from which they were retrieved, and the results were compiled into a single excel spreadsheet. Records obtained through hand searches [38], following best practices and examples for the use of grey literature in WASH meta-analyses [43,44], were also added to this database with a unique numeric identifier.

After removing duplicate titles, each study was screened for its applicability to WASH and WASH systems based on the title and content of the abstract. For a study to be included in the full text review, it had to have an explicit focus on a water, sanitation or hygiene topic or the WASH sector overall. Examples included rural water delivery, fecal sludge management or sector financing. To assist with this screening process, multiple exclusion terms were developed to remove studies that focused on water or wastewater treatment, hydrology, water resource management, and topics bounded more specifically to public health. Of these exclusion criteria, water resource management topics, especially integrated water resource management (IWRM) approaches, had many of the same qualities as systems approaches to WASH. However, after further review, it was determined that the IWRM literature was distinct enough from approaches applied to WASH services that it was prudent to exclude it from the review. Additionally, WRM and IWRM systems approaches have previously been addressed in literature reviews by others [45,46,47].

To ensure that studies were applicable to systems approaches, full text records were also screened to confirm that they referenced two or more factors of the WASH system (e.g., finances, governance, hardware, etc.). While meta-analyses or other reviews were excluded, we retained full texts of these studies to identify other studies cited in the literature to determine if they met the eligibility criteria. If applicable, these studies were included in the ‘hand search’ category. Exclusion terms and criteria for both applicability to WASH and WASH systems are presented in (Appendix A).

#### 2.2.2. Secondary Screening

To accurately characterize the existing landscape of systems approaches to WASH services, the review examined three key attributes of each study that met the primary screening criteria above: method (RQ1), scope (RQ2), and application (RQ3). RQ4, or gaps in knowledge, was addressed through a review of results from all attributes. To assess the qualities of each study in regard to the research questions, a set of study descriptors or important dimensions of the attribute, were developed. Each descriptor represents an important dimension of the attribute and was developed based on comparable literature reviews of WASH and systems-thinking for development topics [35,48] as well as systematic coding procedures outlined by Cooper et al. [38]. Each attribute is described below with descriptors and evaluation criteria presented in a coding form in Appendix A. All secondary screening was conducted by the first author.

##### Method Attribute

The review deductively coded information relevant to the methodology attribute into the following descriptors for each study: methods, data sources, factors, interactions, analytical complexity, and how the method was applied in the study (Table 1). Within each descriptor, we used both inductive and deductive coding techniques to characterize evaluation criteria.

(1)Methods Coding

To capture the range of methods being employed for systems approaches to WASH, we identified the methods for each study by recording both the proper name(s) of the method used by the authors in the text, and inductively coding methods into categories of methodologies. For example, a study that employed a linear regression model would be coded into a ‘statistics’ category while a study that used stock-and-flow models would be coded into a ‘system dynamics’ category. Codes for specific methods were developed for each set of methods that was represented by at least two or more studies. A codebook for the resulting descriptor categories is presented in Appendix A.

(2)Data Sources

To classify the sources of data that the included studies draw on, we coded sources, primarily from data collection descriptions, into common categories including primary and secondary data, interviews, and surveys, among others. Sources of data were coded to best represent relevant categories for each study; for example, one study that described data collection as ‘surveys administered through face to face interviews’ was coded under ‘Surveys’ to represent the primary nature of the data collection.

(3)Factors

To identify the scope of the different elements, aspects, and components of the WASH systems that the included studies evaluated, we openly coded factor categories based on descriptions of variables presented in each study. For example, studies that examined the relationships between willingness to pay and gender of the head of household would be coded as financial and gender, respectively, within the factors descriptor.

(4)Interactions

As factor interaction is a fundamental concept of systems thinking approaches [50,51,52,53], we also evaluated whether or not the study explicitly analyzed interactions or relationships between the factors included in the study. Interaction treatment was evaluated as a yes/no descriptor based directly on the analytical components of the method as described by the authors.

(5)Analytical Complexity

To characterize the complexity of these approaches within the WASH sector, we sought to assess the analytical complexity of each method. As a direct comparison of the multitude and wide variety of anticipated methods did not appear feasible, analytical complexity was evaluated using a low-medium-high rubric based on the degree of specialized knowledge or training required to conduct the analysis as described in Table 1.

(6)Method Application

To evaluate how different methods are applied with each study, we categorized method application into four pre-determined categories (Table 1) ranging from a direct application of an established method (analysis) to a theoretical or conceptual framing of how the WASH topic could be evaluated (approach). A preliminary review of studies prior to the structured literature search revealed that studies could contain both multiple methods and multiple applications of those methods, and thus neither of these criteria were assumed to be mutually exclusive. This resulted in some studies being classified under multiple method applications.

##### Scope Attribute

For RQ2, we evaluated and coded the scope of each study into three study descriptors: aspects of WASH addressed, the focal outcomes of the study, and the contexts in which the approach was applied, if any (Table 2). Taken together, these three coding categories will help to provide more clarity on what aspects of WASH, and in what contexts, systems approaches are being prioritized in the emerging systems literature.

(1)Aspects of WASH

The application of the approach to water, sanitation, and/or hygiene was assessed directly based on the described scope of the study. Categories for this descriptor were not deemed to be mutually exclusive as the scope of the study could include more than one aspect of WASH (i.e., water and sanitation, sanitation and hygiene).

(2)Outcome Scope

A direct comparison of study outcomes and dependent variables was not feasible; thus, we emergently coded outcomes into descriptive categories. For example, a study that focused on promoting good governance and finance for water system operations would be coded as ‘Sustainability’ as it addresses factors related to the ongoing operation of an existing water delivery scheme. Conversely, studies which focused on the sale of new latrines would be coded as expanding ‘Access’ to services.

(3)Context

Context was coded using two a priori, deductive coding categories that assessed context based on both a rural/urban classification and by administrative designations of community, city, regional (including district), and national. A ‘Sector’ category was also included for studies which focused on sector-wide issues across multiple geographies.

##### Application Attribute

The third attribute examines the application of the study to a WASH program or project, including the phase of project implementation, the geographic location, as well as any reported impacts that occurred as a result of the study (Table 3).

(1)Project Application

The project application descriptor was developed to assess what aspects of WASH projects systems approaches are being used to, or proposed to, investigate. We evaluated project application based on a priori codes of project phases including planning (before), implementation (during), and evaluation (after). Studies that evaluated WASH services or factors to support services but were not specific to any aspect of an implementation project were classified as case studies. Any studies that were not related to a specific project were labeled as none. Similar to the method application, the project application descriptor was not assumed to be mutually exclusive. For example, a project that evaluated factors that were critical to project implementation, and drew on multiple cases, would be categorized as both implementation and case studies.

(2)Location

To assess where systems approaches are being implemented, we recorded the specific location(s) of each study, where applicable, to assess the geographic distribution of study locations. Studies did not need to be integral to project implementation in order to be assessed for the location descriptor. If a study specified that work was conducted in multiple districts or regions within a country, the count and names of the locations were recorded as well. In classifying locations, we used the United Nations geoscheme for regions and sub-regions globally [54].

(3)Reporting of Impacts

To address RQ3 (What evidence exists for systems approaches?) we sought to capture all reporting of impacts on services or other factors of the WASH system (e.g., policy, behavior change) that occurred due to the use of the analysis, tool, framework or approach implemented in the study. In collecting information on study impacts, we did not seek to assess the quality of evidence presented in reporting those impacts. Instead, we recorded whether or not impacts were reported, and captured the descriptions of the impacts as described by the authors.

In addition to the three study attributes and descriptors defined above, the review also captured report characteristics for each study, including author(s), journal, year published, whether the study represented a peer-reviewed, grey or organizational literature source, and if the study was available as open-access.

## 3. Results

### 3.1. Literature Search Results

Through our search, we obtained 8139 articles from databases searches and 100 from hand searches that were potentially eligible for the review. Of these articles, 472 were eliminated because they were duplicates, resulting in 7767 articles eligible for primary screening (Figure 1). After reviewing articles for their applicability to WASH and treatment of systems as described in Section 2.1, 584 full text articles were obtained for secondary screening to determine if enough information was available on our study’s descriptors. Of these articles, 451 were excluded because they either did not have sufficient information to address the study descriptors (e.g., research briefs from grey literature) or because the full text review indicated they did not meet the initial screening criteria. This resulted in 133 studies included in the final review. A majority of the studies (75) were classified as peer-reviewed literature, 35 were classified as grey literature, and 23 as organizational literature. Full results of the review, including a breakdown of descriptors by literature type, is presented in Appendix A. Unless otherwise specified, all percentages presented in the following sections are expressed in regard to the total count of studies (*n* = 133).

### 3.2. Method Assessment (RQ1)

Overall, we identified 41 unique methods in the peer-reviewed, grey, and organizational studies. These methods could be employed non-exclusively (i.e., papers could be categorized to more than one method). Of these methods, unique frameworks (e.g., sanitation cityscape conceptual framework [55]), composite scoring, qualitative data analysis, and statistical methods were employed most frequently (Figure 2). A breakdown of methods by literature source indicates that each of the top three methods (frameworks, composite scoring, and qualitative data analysis) were most prevalent in the grey and organizational literature; whereas a majority of the statistical, social network analysis, system dynamics, and causal loop diagraming methods were the most prevalent methods applied in the peer-reviewed literature.

When the studies were assessed for analytical complexity, results indicated a relatively even distribution of high complexity (38%) (e.g., statistics, network analysis, system dynamics), medium complexity (37%) (e.g., qualitative data analysis, composite scoring), and low complexity (25%) (e.g., frameworks, checklists). Surprisingly for a review of systems approaches, only 35 studies (26%) included a method that explicitly analyzed interactions between factors that were considered in the analysis (e.g., causal loop diagraming, network analysis). Of these, 26 (81%) were from the peer-reviewed literature.

Our review of sources of data, another non-exclusive descriptor, indicated that secondary data (53%) (e.g., government reports, national surveys, etc.) and interviews (47%) were the two most common sources. These were followed by primary data (28%), observations (276%), surveys (24%), focus groups (23%), expert opinions (21%), and workshops (5%). Notably, secondary data (e.g., national surveys, document review) were used in a majority (83%) of studies in the organizational literature, the most of any data source by literature type.

Forty unique factors were referenced by at least two or more studies from our factor coding (Figure 3). Of these, *Financial* (74%) was overwhelmingly the most common factor, followed by *Technical* (53%), *Institutional* (43%), *Social* (41%), and *Environmental* (40%) factors. However, these factors were not equally studied across the different literature types; *Financial* factors (e.g., tariffs) were nearly ubiquitous in the organizational (91%) and grey (89%) literature than they were in the peer-reviewed literature (61%). Conversely, *Economic* factors (e.g., markets) were twice as frequently in the peer-reviewed literature (40%) than the grey or organizational literature (both 17%). Across all literature sources, the average number of factors each study examined was 7.9, with the most common count of factors being five (17%), six (15%), eight (14%), and seven (11%) factors, in descending order of frequency.

Our assessment of method application, a non-exclusive category (multiple descriptors could apply to a study), indicated that methods were used nearly equally as tools (35%), frameworks (34%), analyses (32%), and approaches (26%). However, as with methods and factors, notable differences exist across the literature sources, with nearly half of all studies from the organizational literature proposing a novel tool (52%), and a smaller proportion of grey and organizational studies presenting a novel framework of 40% and 43%, respectively. In the peer-reviewed literature, we observed that a large proportion of studies (39%) represented methods used for one-off case study analyses.

### 3.3. Scope Assessment (RQ2)

Results of coding the scope of the studies showed that the most focused on or were intended for, were eater services (41%) followed by sanitation (22%), and then eater and sanitation together (15%). Few studies explored issues around hygiene (2%) or sanitation and hygiene (3%), while a notably larger proportion examined WASH-sector topics at large (12%). Despite open-coding outcomes in the review, our assessment identified only two main themes in the literature: service sustainability (78%) and access to services (22%). Other focal areas such as improving health outcomes, promoting behavior change, fostering coordination, improving functionality, and advocating for policy change, were each mentioned in less than 3% of the studies. These results highlight the prominence of sustainability as a key focal outcome, with 104 of the 133 studies addressing a service sustainability issue.

Another non-exclusive descriptor of the scope attribute, context, showed a high propensity for rural (74%) and community-focused (59%) study scopes. Urban contexts were studied notably less (43%), but 23% of studies purported to be applicable to both rural and urban contexts. However, only 11% of the studies focused solely on city-level scope, highlighting the traditional lack of attention paid to peri-urban contexts. Many studies were focused at a regional level (32%), reflecting current trends of district-wide approaches in the WASH sector, while fewer were focused on national-level systems (20%). Only 8% of the studies focused purely on sector-level issues.

### 3.4. Application Assessment (RQ3)

Results for project application showed that a majority of studies included in the review were focused on evaluating project outcomes (42%), while 30% focused on planning, and only 11% on implementation. While an overwhelming majority of studies had some form of project application (116 of 133), results indicated that 35% of all studies represented case studies that focused on only one geographic context, which may limit the generalizability of these studies. Studies that included a project application were mostly equally distributed between countries in East Africa (40%), South Asia (32%), and West Africa (24%). The review also showed the disproportionate focus of studies that assessed projects in India (15%), Ghana (14%), and Uganda (13%). Overall, applications in 60 countries were represented in the literature.

Our evaluation of study impacts also showed that 32 of the 133 studies (24%) reported some form of impacts that resulted because of the use of the analysis, tool, framework or approach. Non-exclusive open coding of the study impacts identified eight types of reported impacts, including: uptake of the tool, framework or approach (12%); effects on services (7%); policy changes (6%); improvements in coordination (4%); behavior change (3%); financial impacts (2%;, impacts on users (2%); health impacts (>1%); and changes in levels of access to services (>1%).

## 4. Discussion

Results of the review paint a complex scene of the existing landscape of analyses, tools, frameworks, and approaches that are incorporated into systems approaches for WASH. In this section, we reflect on some of the most prominent themes that emerged from the analysis as they relate to methods employed (RQ1), context and application (RQ2), and evidence of impact (RQ3). In addition, we discuss gaps in the existing knowledge base for systems approaches to WASH (RQ4). In particular we highlight findings salient to WASH sector practitioners, including analytical complexity (RQ1–RQ4) and focus and scope (RQ1–RQ4). In so doing, we offer insights and suggestions for future research and practice in systems approaches to WASH.

### 4.1. Many Methods, More Factors, and a Myriad of Frameworks

While we identified a wide range of methods in the systems approaches literature, a majority of these methods appear to require a modicum of specific analytical capacity in order to implement. Overall, 75% of studies across all literature sources were coded as having a high or medium analytical complexity to conduct. Further, nearly a quarter of the ‘high’ and ‘medium’ complexity tools were applied at the mostly rural household or community level. This finding, in part, is due to the large proportion of peer-reviewed studies in the review (75 of 133), however it highlights a growing disconnect between the increasing complexity of managing WASH services in urban environments and the focus of the literature on contexts with lower population densities. The studies where more accessible approaches were proposed (e.g., frameworks, approaches) were generally vague on how these approaches can be put into practice at the project or program level.

The review also highlighted a wide diversity of factors used across the literature to examine WASH systems. In addition to traditional technical, financial, and environmental aspects of WASH service sustainability, we observed many references to factors focused on social, institutional, community, and user aspects of WASH systems. However, despite the diversity of methods and factors, few studies explicitly considered the interactions between these factors. Studies which did explore interactions (*n* = 35) were almost exclusively from peer-reviewed literature (*n* = 26), suggesting that there is less focus on interactions in the practitioner-focused grey or organizational literature. Moreover, of the studies that did investigate interactions, only four reported any impacts that occurred because of the use of the tool, framework or approach. Considering that interactions are a fundamental component of systems thinking [50,56,57], their omission from many of the analytical methods, and the lack of reported impacts in studies where they were included, suggests that current work may be neglecting a key dimension of systems approaches for WASH. This finding tracked closely with analytical complexity where 20% of methods with high analytical complexity included some analysis of interactions, whereas medium and low complexity methods only accounted for interactions in 5% and 2% of studies, respectively. Additionally, we found little mention in the literature regarding descriptions of direct versus indirect effects, feedback mechanisms or dynamic behavior, key concepts of complex systems thinking [52].

Additionally, within the WASH systems literature, one-third of all studies (*n* = 41) represented an original/novel framework. These frameworks were proportionally more common in the grey and organizational literature. We find this notable as others reported that 80% of WASH sector professionals have indicated in previous studies that they did not use a formal planning tool in the implementation of water and sanitation projects because “the context-specific nature of project planning decreases the applicability of a planning framework” [58] (p. 79). On reflection, this raises questions about the demand for, development of, and application of, new frameworks for evaluating WASH systems, as well as the ability to compare findings across frameworks and contexts to learn and advance the sector. Contrary to how many of the studies present these tools, frameworks, and approaches, the review indicates that there is a lack of evidence for their broad application across diverse settings. For more information on individual frameworks presented by various studies see Appendix A for source material.

### 4.2. Geographic Dispersion of Applications

Overall, 116 of the 133 studies referenced at least one application of the tool, however, applications were not equally dispersed across geographies. In particular, applications in either India, Ghana or Uganda, were referenced in 41% of the studies reviewed. Our findings align with findings from previous studies on the disproportionate focus of the international development sector in African [59] and Southeast Asian [60] countries in general, and may limit the generalizability of systems approaches to WASH, where many tools, frameworks, and approaches appear to be built specifically to conditions in these contexts. There are some noteworthy exceptions to this trend where systems approaches have been informed by findings from globally dispersed contexts. For example, one study representing a “Suite of Tools” for a systems-based approach to water service delivery [61] draws on the authors’ experience working in 30 districts across nine countries and offers recommendations on how the tools can be used to assess different aspects of factors supporting service delivery. Similarly, another study [62] presents a novel framework for promoting market-based approaches to sanitation based on experiences from 22 World Bank projects across 13 countries. Considering the important role that local context plays in influencing WASH service delivery models and sustainability [63], the limited geographic focus within and across studies included in this review raises questions about the external validity of findings and insights within the existing body of knowledge.

### 4.3. Reporting on Service Sustainability

As described above, one motivation for conducting this review is the hypothesis that an increase in systems approaches to WASH predicated their intention to address pervasive issues of service sustainability. Our findings for outcome scope showed that while sustainability is indeed the main focal area of many systems approaches, only 24 of the 104 studies (23%) that focused on service sustainability included any mention of study impacts. While assessments of service sustainability are relatively new to the sector and require some form of longitudinal data collection, there is no shortage of tools for measuring service delivery outcomes in WASH that would hinder reporting on these dimensions in contemporary literature [64]. Further, while reporting impacts that result from a study (as opposed to findings) is not common practice in academic journals, half of the studies (16 of 32) that cited some impact were from peer-reviewed literature.

Of the reported impacts, some identified null or undesired outcomes such as a lack of change in hygiene practices [65,66] or unmaterialized changes in national policy [67]. These studies represent an encouraging and necessary trend that has been embraced by the sector [68], yet more reporting and dissemination of programmatic outcomes, both positive and negative, is clearly needed. Overall, the low incidence of studies reporting on impacts on WASH services (7%) presents a challenge for developing the evidence base for the effect that systems approaches have directly on the sustainability of WASH services. Thus, in order to determine the effectiveness of these approaches, more information is required that connects the implementation of these approaches to tangible service delivery outcomes, and ultimately, improvements in public health.

### 4.4. Reporting on Resources and Replicability

The review also highlights the limited potential for the replicability, or scalability, of many of the approaches proposed in the literature. For example, one study that was classified as both a tool and framework [69] consisted of data collection through focus groups and interviews—used to score multi-criteria decision analysis matrices using an analytical hierarchy process—followed by a multi-step statistical analysis and social network analysis. While applying this framework may produce rich insights into a particular context, the approach overall may be difficult to replicate due to resource needs that include external, highly skilled WASH professionals. Although we did not conduct a direct assessment of the replicability or scalability of each approach, we likely would have been hindered in doing so, since few studies, with some noted exceptions [37,70,71], reported on the resources that are required to implement the proposed approach. In order to better evaluate the practicality of these approaches, more reporting is needed on the resources required for studies which are applied across multiple contexts.

### 4.5. Diversifying Methods, Analytical Complexity and WASH Scope

While the review highlighted the wide variety of methods being employed in systems approaches to WASH, a majority of these methods were of a medium or high analytical complexity applied in only one, usually local, context or none at all. The result of this combination is that the largest proportion of studies included in the review appears to use complex analytical methods to assess an issue at the local-services level (e.g., household, community) (Figure 4). Examples of this genre of systems approaches include evaluating contextual, technical, social, and financial detriments of handpump functionality through Bayesian network analysis [72], using agent-based modeling to assess household water quality [73], and the application of fuzzy set qualitative comparative analysis for assessing failure in community sanitation projects [74]. This trend suggests that few of the methods intended for addressing local service sustainability issues are likely accessible to those who are tasked with making critical decisions at the local level around WASH service provision.

The strong preference of systems approaches to focus on rural water services (48% of all studies) also points to another significant knowledge gap in the existing literature. Although water services are an essential part of WASH that need to be sustained in order to deliver public health impacts, recent studies of environmental enteropathy effects [75,76,77] have shown that there is little impact from water services alone if sanitation and hygiene practices are not also improved and sustained simultaneously. Furthermore, as a majority of the world’s population now resides in urban environments, and these areas will continue to grow exponentially in population and density [78], systems approaches need to begin shifting their focus to these more complex contexts in order to meet the needs of future populations. As such, it would be in the WASH sector’s interest to begin expanding systems approaches beyond a traditional focus on rural water service delivery to include a focused analysis of the how the multitude of factors in peri-urban or informal settlements interact with one another to inhibit or promote WASH services.

Overall, while our review identified a large number of studies with a wide variety of methods, scopes, and project applications, we found that studies generally tended to represent one-off case studies (41%) that employed a method requiring medium-to-high analytical complexity (75%), most commonly applied to the analysis of a rural water context (48%), that generally focused on financial (75%) or technical factors (54%), and did not explicitly consider interactions between factors (75%).

### 4.6. Limitations

By the nature of such a large secondary data review, there are inherent limitations to the findings presented in this study. For example, while we sought to gather as much applicable literature as possible, it is possible that some grey or organizational literature may not have been captured in the review and thus may be underrepresented in the results. As with any literature review, our choice of screening criteria and principal categories focused our analysis; different screening criteria and categories would alter the results.

## 5. Conclusions

We conducted a systematic review of peer-reviewed, grey, and organizational literature of systems approaches to WASH. Our objectives were to assess the methods and factors being included in systems approaches, the contextual scope of these approaches, their geographic and programmatic applications, evidence of their impact on WASH services, and gaps in the existing knowledge base. From 7764 unique search results we identified 130 studies that met our two-phase screening criteria and evaluated these articles for 12 study descriptors across three categories of method, scope, and application. We then evaluated needs and gaps based upon this review.

The results of the review indicate a propensity in the systems approaches literature towards the application of complex analytical methods to singular case studies focused on financial and technical factors and are most commonly applied to rural water service delivery contexts. Despite finding a large diversity of factors in the literature, few studies explicitly evaluated interactions or relationships between factors, a fundamental concept of systems thinking. The review also found inadequate and inconclusive information to assess the impacts—positive, null or negative—that systems approaches have demonstrated on WASH service sustainability. While this could be related to the time needed to foster systems change and impacts, increased reporting and a plan for monitoring and reporting impacts over time are needed. Thus, our key recommendations call for: (i) a diversification of the methods, scopes, and applications of systems approaches for WASH; (ii) further investigation and application of system approaches that explicitly consider factor interactions; (iii) increased reporting of resources required to implement the approaches; and (iv) more documentation of the impacts to WASH services that result from the application of a systems analysis, tool, framework or approach. Overall, these findings provide a robust survey of the existing landscape of systems approaches for WASH and illuminate a path for future research in this emerging field.

## Figures and Tables

**Figure 1 ijerph-17-00702-f001:**
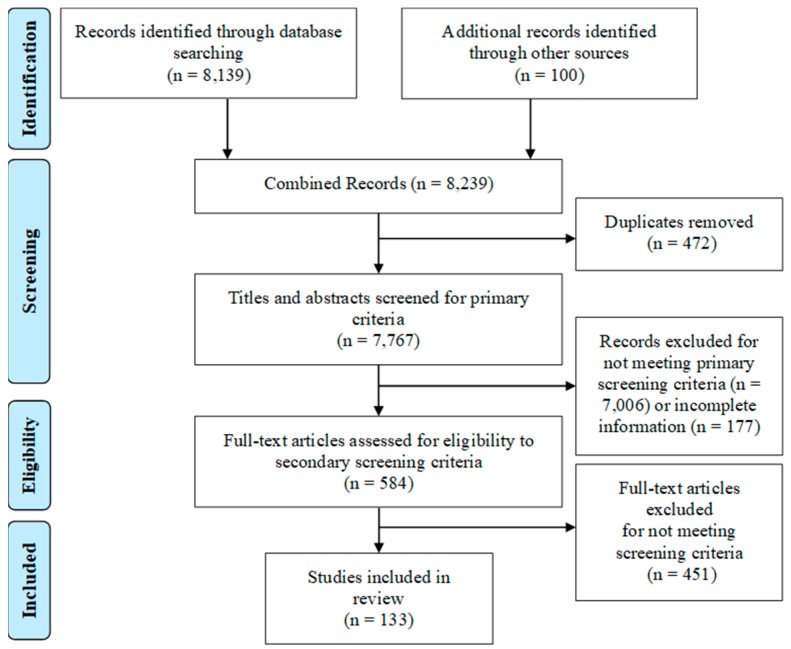
Preferred Reporting Items for Systematic Literature Reviews and Meta-Analyses (PRISMA) flowchart with selection of articles included in this review.

**Figure 2 ijerph-17-00702-f002:**
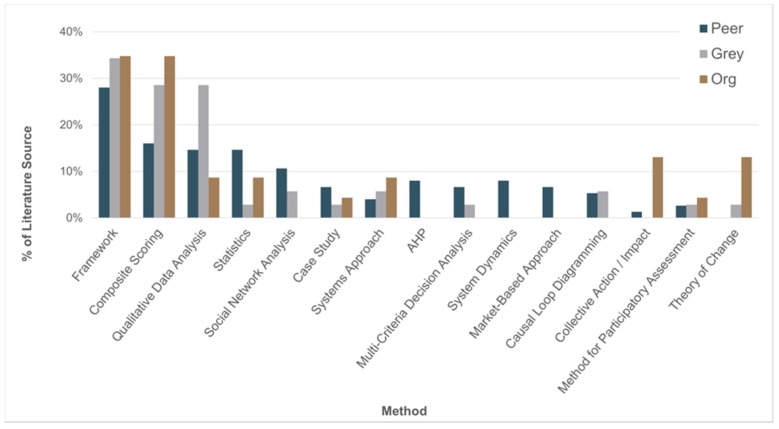
Relative frequency of the 15 most frequently used methods by literature source, listed in descending order of frequency of total references from all literature sources (peer-reviewed = 75; grey = 35, organizational (org) = 23).

**Figure 3 ijerph-17-00702-f003:**
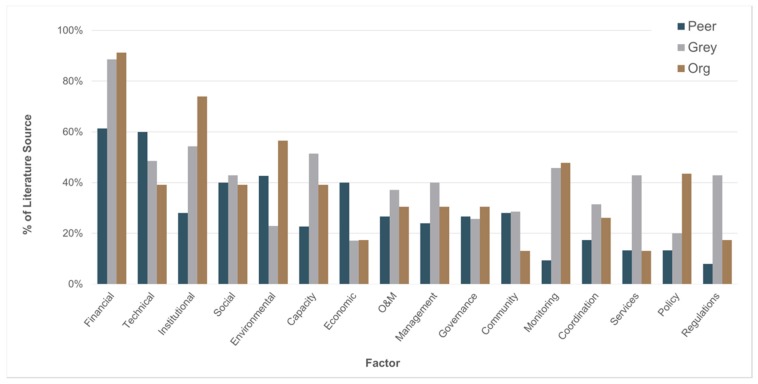
Relative frequency of factors referenced in at least 20% of all studies listed in descending order of frequency of total references from all literature sources (peer-reviewed = 75; grey = 35; organizational (org) = 23).

**Figure 4 ijerph-17-00702-f004:**
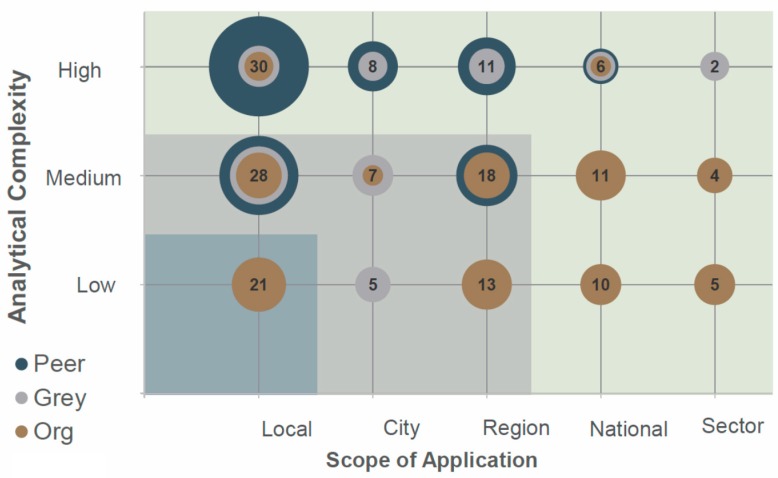
Analytical complexity versus scope of application by literature source.

**Table 1 ijerph-17-00702-t001:** Descriptors and criteria for assessing the method attribute.

Method Descriptors	Criteria
**Method ^2^**	Stated name of method of analysis or approach employed by the study (open response) and open coding of methodologies.
**Data Sources ^2^**	Open coding of source of data for the study (e.g., surveys, interviews, focus groups, observations, etc.).
**Factors ^2^**	Open coding of factors included in the study and count (e.g., finance, hardware, regulations).
**Interactions ^1^**	Evaluates if the study considers interactions among factors (coded as yes/no).
**Analytical Complexity ^1^**	Low: Non-computational tasks can be completed without specialized knowledge or training. Medium: Some specific knowledge or training is required to complete computational tasks. High: Process requires a high level of specialized knowledge or training to conduct the analysis.
**Method Application ^1^**	Analysis: Application of an established analytical method, presented without a broader theory on how it should be applied, or steps for applying it (e.g., statistical regression of survey data). Tool: A discrete, standalone activity or analysis presented with sufficient detail to be readily replicated, or an analytical program or software (e.g., checklist for assessing service sustainability). Framework: A guiding outline of activities, analyses or procedures for applying the method (e.g., list of principles for engaging in local systems [49]). Approach: A theoretical or conceptual construct, without discrete steps for implementation of the method (e.g., collective impact/action).

^1^ a priori, deductive criteria; ^2^ open or emergent criteria.

**Table 2 ijerph-17-00702-t002:** Descriptors criteria for assessing the scope attribute.

Scope Descriptors	Criteria
**Aspects of WASH ^1^**	Water, sanitation, and/or hygiene
**Outcome Scope ^2^**	Emergent coding of study outcomes or dependent variable (e.g., behavior change, access to services, service sustainability, etc.)
**Context ^1^**	Rural/urban
	Local (including community), city, regional (including district), national sector


^1^ a priori, deductive criteria; ^2^ open or emergent criteria.

**Table 3 ijerph-17-00702-t003:** Descriptors and criteria for assessing the application attribute.

Application Descriptors	Criteria
**Project Application ^1^**	Assessment of what stage, if any, the study was implemented in relation to a WASH infrastructure or services project: Planning—Used to plan future WASH service delivery project; Implementation—Method used to guide the implementation of a WASH project; Evaluation—Post-project assessment of outcomes; Case Study—Standalone analysis of a specific case(s), not related to a WASH project; None—No project application conducted.
**Location ^1^**	Name and count of geographic locations where study was applied (Based on UN Geoscheme [54].
**Reporting of Impacts ^2^**	Stated outcomes or effects on WASH services that occurred as a result of the application of the study (if any).

^1^ a priori, deductive criteria; ^2^ open or emergent criteria.

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
