# Peer review of "System Approaches to Water, Sanitation, and Hygiene: A Systematic Literature Review"

_ijerph, 2020, doi:10.3390/ijerph17030702_

Round 1

Reviewer 1 Report

A large amount of effort has gone into the manuscript, and it covers an extremely complex set of concepts and parameters. 

The material/research is broad and complex, but the manuscript is well written and pulls together relevant findings in an organized manner. I agree that the work is a robust survey of the existing landscape and clarifies the paths forward for future WASH-related research.

The systematic process of evaluating article content and quality appear to be excellent, which yielded a solid set of articles to actually use for final review; a major effort was required.

Line 316 is not quite clear to me - is a word missing? I did not note anything else relevant to language and style.

Author Response

Comment

Line 316 is not quite clear to me - is a word missing? I did not note anything else relevant to language and style.

Response

Thank you for noting this. This sentence has been revised to remove an additional word which made the sentence incomplete.

Reviewer 2 Report

This paper studied systems approaches to water, sanitation and hygiene. I think it is very interesting. However, I have a few queries for reply. Firslty, as a review paper, I think more literatures should be analyzed and cite no more than 100 literatures. Secondly, the authors proposed four recommendations for improving the evidence. But how to verity its effectiveness and praticability? The author should illustrate more clearly. Finally, some tables described not clearly, for example table 1 and 2. Please demonstrat more clearly.

Author Response

Comment

Firstly, as a review paper, I think more literatures should be analyzed and cite no more than 100 literatures.

Response

While the reviewer has a point, in our manuscript we did not develop, nor are we aware of, any criteria for limiting the total number of studies in a systematic literature review or meta-analysis. Because our study focused on an emerging body of knowledge spread across diverse literature sources, we anticipated that a larger number of studies would be included in the review than is common to literature reviews that are narrower in scope. Further, a recent comparative systematic literature review of peer and grey literature on community-led total sanitation (Venkataramanan 2018, see below) similarly identified a large number of studies (n=200). Thus, we believe that the number of studies included in the final analysis is appropriate because the topic of systems approaches to WASH– as we have defined it – is inherently broad in scope and all of the studies included met our predetermined screening criterion.

Venkataramanan, V.; Crocker, J.; Karon, A.; Bartram, J. Community-Led Total Sanitation: A Mixed-Methods Systematic Review of Evidence and Its Quality. Environmental Health Perspectives 2018, 126.

Comment

Secondly, the authors proposed four recommendations for improving the evidence. But how to verify its effectiveness and praticability? The author should illustrate more clearly.

Response

In both the introductory sections and the concluding sections we discuss the need to evaluate the implementation of systems approaches to WASH to tangible service delivery outcomes, and ultimately improvements in public health. Our results show that only 7% reported on impacts to service sustainability that resulted from applying  the systems approach in the study (Section 3.4, Application Assessment RQ3). We conclude that this very low occurrence of reporting on service delivery outcomes is insufficient to evaluate the effectiveness of systems approaches. Thus, we posit that a more ‘effective’ approach will result in an improvement in the sustainability of WASH services within a given context.  

In terms of practicality, which we framed as ‘replicability’ in our study, we similarly observe and note in our findings that a large portion of the studies were only applied in one context, which we concluded limit both their external validity and generalizability. Thus, because many studies represent one-off experiments, we concluded that there is currently insufficient information available in the literature to evaluate the replicability (practicality) of systems approaches for WASH. Thus, at a minimum, we need a larger body of work applied across multiple contexts in order to have sufficient information to assess the practical application of these approaches. We have added a clarifying statement to the discussion section (Lines 524-526).

Comment

Finally, some tables described not clearly, for example table 1 and 2. Please demonstrate more clearly.

Response

Thank you for highlighting this issue. We have revised all the table captions to clarify that each of Tables 1-3 present descriptors (i.e. dimensions) and criteria for assessing each attribute.

Is this an addition to the paper?  If so, best practice is to indicate the addition in the responses to the reviewers.

Yes. Added a summary section at the top of this response. Let me know if this is a good format for highlighting those changes to the text.

Reviewer 3 Report

Thank you for the opportunity to review this article. It is an important contribution to the discourse around systems approaches to WASH, and a good overview of some of the key literature related to the topic.  

The paper presents a list of frameworks and approaches, but little detail or explanation on them, so the article feels lacking in robust content, and more focused on the methodology and identification of gaps (findings from the analysis). This section in particular should be elaborated upon: 

"This systems perspective of WASH has been embraced by many in the sector and has led to a 56 rapid expansion in the number of tools, frameworks and approaches for understanding the 57 interconnected factors of WASH services[26] including; collective action coalitions [27]; multi-58 criteria decision analysis [28]; market-based approaches [29], soft systems methodologies [30], 59 composite scoring [31], and system dynamics [32], among others."

The paper could be improved by including relevant chapters from the following book: 

Neely, K (Ed) Systems Thinking and WASH: 

https://www.amazon.com/Systems-Thinking-WASH-Kate-Neely/dp/1788530268

This book has the same premise as your article, in terms of sustainability and achieving SDG6: 

"Water supplies in developing countries fail at unacceptable rates. In an era of high technology and a global drive for sustainable water and sanitation (SDG6), we need to find solutions to the wicked problems that characterize water for development programmes around the world. Systems Thinking and WASH introduces practitioners, researchers, programme managers and donors to the tools and approaches that have been most successful in this area. This book explores the different applications of systems thinking used by an interdisciplinary group of WASH researchers and practitioners. With additional commentary from the field, each chapter helps us to imagine different ways to understand and work with communities, development agencies and governments to create a better world through more appropriate WASH (water, sanitation and hygiene) programming. The book includes an annotated list of additional resources that anyone interested in non-linearity, complex adaptive systems, systems thinking, social network analysis or system dynamics will find useful as a practical guide to getting started. This book is highly important reading for WASH programme managers, government and NGO staff and donor agencies interested in the application of systems thinking techniques."

Especially the soft systems thinking tools elaborated in the book would be good to mention, in light of the following RQ. 

'What 80 methods (i.e. analyses) are being employed for understanding and engaging with WASH systems?'

If this book and the chapters within it did not meet the screening criteria, could you please explain why. 

In terms of the argument about 

" the generalizability of systems approaches to WASH" (line 470) - I think more can be talked about in terms of the way that systems approaches highlight context and a deep understanding of the unique factors that influence a system. This is in contract to any "generalizability" aims, despite there being some examples of tools and approaches that have been used in a range of contexts. Also, by mentioning these approaches explicitly and naming the authors (rather than numbering the source as per the rest of the paper), the paper reads as though it is commending these generalised approaches over more contextualised ones. 

Minor edits:

Overall the language is clear and easy to follow. 

Line 458: In reflection should be On reflection 

Line 518: that are required (extra space appears)

Congratulations on a sound first draft of the paper. 

Author Response

Comment

The paper presents a list of frameworks and approaches, but little detail or explanation on them, so the article feels lacking in robust content, and more focused on the methodology and identification of gaps (findings from the analysis). This section in particular should be elaborated upon: 

"This systems perspective of WASH has been embraced by many in the sector and has led to a rapid expansion in the number of tools, frameworks and approaches for understanding the interconnected factors of WASH services [26] including; collective action coalitions [27]; multi-criteria decision analysis [28]; market-based approaches [29], soft systems methodologies [30], composite scoring [31], and system dynamics [32], among others."

Response

Thank you for the comment. In the introduction of the manuscript we present one of the primary purposes for the literature review is to identify gaps in the existing knowledge base, as is commonly the motivation behind a systematic review of literature on a given topic. Thus, the stated analytical objectives of the literature search is to quantify and report on important attributes of the studies included in the review, as opposed to comparing the relative merits of study’s approach against one another. In regard to frameworks in particular, we note in the conclusion that the review identified 44 studies which presented unique or novel frameworks (Lines 456-457). Given the scope and limitations of the manuscript, we do not believe a comparison of this large number of frameworks is appropriate, nor is a stated objective of the study.

The passage referenced in this comment from the introduction is intended to serve an illustrative point that highlights the diversity of methods in the field as a justification for the need of systematic literature review.

For our readers who are interested in more of the details of the unique frameworks found in the review, we have added a sentence in the Discussion section (Lines 470-471) pointing them to Table SI 3-1 which contains the source materials through which they can find these frameworks.

Comment

The paper could be improved by including relevant chapters from the following book:

Neely, K (Ed) Systems Thinking and WASH:

https://www.amazon.com/Systems-Thinking-WASH-Kate-Neely/dp/1788530268

If this book and the chapters within it did not meet the screening criteria, could you please explain why.

Response

Thank you.  We believe the book by  K Neely is excellent and provides complementary information to the work presented in our study. The book was not identified in any of the databases included in the literature search (SI 2); however, we did review an early copy of the chapters for inclusion as a possible hand search. Upon re-review, we have determined that three of the chapters of the book do meet the screening criteria for the study and thus have been included in the database and final analysis. While the addition of these studies did affect the numerical figures presented in the manuscript, they did alter any of the results, conclusions or commentary in the discussion section.

Comment

In terms of the argument about  "the generalizability of systems approaches to WASH" (line 470) - I think more can be talked about in terms of the way that systems approaches highlight context and a deep understanding of the unique factors that influence a system. This is in contrast to any "generalizability" aims, despite there being some examples of tools and approaches that have been used in a range of contexts. Also, by mentioning these approaches explicitly and naming the authors (rather than numbering the source as per the rest of the paper), the paper reads as though it is commending these generalised approaches over more contextualised ones. 

Response

Thank you for this insightful comment. While we firmly agree that ‘context is key’ in understanding local systems, this dimension arises as an end product of the process of using a systems approach rather than being an attribute of the approach itself. In this study, we are concerned with the generalizability of the method, not the findings of the method in a given context. Thus, an effective and practical systems approach for WASH would be one that has been implemented in multiple different contexts and has been shown to faithfully represent the uniqueness of each one of those contexts. If a particular systems approach is built around a specific context and is only applicable to that context than it is not of much utility to others in the sector working in other contexts. These approaches would be by their nature ad-hoc and limited in their scalability and application. Considering the current scale of issues with the delivery of sustainable WASH services worldwide (see 1. Introduction), and our understanding of how knowledge is disseminated through the WASH sector, we do not believe that ad-hoc approaches to WASH are valued by practitioners or professionals.

In response to the second part of this comment, we have removed references to specific authors names in the discussion section you referenced so as not to appear to infer an biases about the authors.

Comment 

Line 458: In reflection should be On reflection

Response

This line has been revised to read “On reflection…”

Comment

Line 518: that are required (extra space appears)

Response

The extra space has been deleted in this line.

Round 2

Reviewer 2 Report

The revised version has addressed all my concerns.